# Composition of Prokaryotic and Eukaryotic Microbial Communities in Waters around the Florida Reef Tract

**DOI:** 10.3390/microorganisms9061120

**Published:** 2021-05-21

**Authors:** Peeter Laas, Kelly Ugarelli, Michael Absten, Breege Boyer, Henry Briceño, Ulrich Stingl

**Affiliations:** 1Fort Lauderdale Research & Education Center, Department of Microbiology & Cell Science, Institute for Food and Agricultural Sciences (IFAS), University of Florida, Davie, FL 33314, USA; peeter.laas@ufl.edu (P.L.); kugarelli@ufl.edu (K.U.); 2Institute of the Environment, Florida International University, Miami, FL 33199, USA; abstenj@fiu.edu (M.A.); bboyer@fiu.edu (B.B.); bricenoh@fiu.edu (H.B.)

**Keywords:** marine microbial communities, reef water, Florida Reef Tract, water quality

## Abstract

The Florida Keys, a delicate archipelago of sub-tropical islands extending from the south-eastern tip of Florida, host the vast majority of the only coral barrier reef in the continental United States. Abiotic as well as microbial components of the surrounding waters are pivotal for the health of reef habitats, and thus could play an important role in understanding the development and transmission of coral diseases in Florida. In this study, we analyzed microbial community structure and abiotic factors in waters around the Florida Reef Tract. Both bacterial and eukaryotic community structure were significantly linked with variations in temperature, dissolved oxygen, and total organic carbon values. High abundances of copiotrophic bacteria as well as several potentially harmful microbes, including coral pathogens, fish parasites and taxa that have been previously associated with Red Tide and shellfish poisoning were present in our datasets and may have a pivotal impact on reef health in this ecosystem.

## 1. Introduction

Coral reefs around the Florida Keys constitute the main part of the third largest barrier reef ecosystem in the world [1]. In addition to the reefs, the area around the Keys is comprised of diverse habitats such as shallow seagrass meadows and mangrove forests. These ecosystems are constantly threatened by global climate change (e.g., ocean warming and ocean acidification), human activities (e.g., fishing and pollution), hurricanes, and tropical storms. In 1990, the Florida Keys National Marine Sanctuary (FKNMS) was established to protect the only coral barrier reef in the continental United States, which provides essential ecosystem services and represents a very important source of food and income for coastal communities [2]. FKNMS annually attracts nearly five million visitors who collectively contribute to its $4.4 billion economic value (data from 2017; https://marinesanctuary.org/wp-content/uploads/2019/07/FKNMS-Report-Final-072819.pdf, accessed on 11 May 2021) through marine-related activities in the sanctuary, including fishing, snorkeling, diving, wildlife viewing, boating and other activities.

The Florida Keys Reef Tract has experienced several major disease outbreaks over the past four decades that have drastically changed the reef ecosystems [3]. Therefore, the preservation of the Florida Keys has become a national priority in the USA [4] and unprecedented restoration efforts are on the way to restore parts of the nearly 90% of original coral cover that was lost (https://www.fisheries.noaa.gov/southeast/habitat-conservation/restoring-seven-iconic-reefs-mission-recover-coral-reefs-florida-keys, accessed on 11 May 2021).

Most recently, stony coral tissue loss disease (SCTLD), which was first identified in 2014 off the coast of Virginia Key, has affected at least 23 reef-building coral species, especially on the outer reef parts [5,6,7,8]. The disease often results in whole colony mortality [5,9,10]. Aquaria studies have shown that disease transmission can occur through direct contact and through the water column. Additionally, disease lesions are significantly impacted (stopped or slowed) by antibiotic treatment, indicating a bacterial origin of the disease [7]. However, thus far, no single pathogen has been identified as the cause of this outbreak.

Coral reefs and their well-structured associated microbial communities are extremely complex and should be seen as parts of an ecosystem with a strong benthic–pelagic exchange [11,12,13]. Therefore, the impact of abiotic and biotic components of reef waters on corals and coral health cannot be overestimated. While presumably not related to the current SCTLD outbreak in Florida, ocean warming, pH decrease, overfishing and coastal pollution are the main threats to coral reefs worldwide [14,15,16]. Increases in sea surface temperatures cause coral bleaching, which is recognized as one of the main concerns over the coming decades; however, an even greater threat can arise from the increasing frequency and impact of coral diseases [17]. The increased prevalence of potential coral diseases is presumed to be driven by nutrient enrichment in nearshore waters [14,18] and is usually also correlated to higher temperatures and increased total suspended solids (TSS; [19]). The FKNMS is directly influenced by water masses with distinct nutrient content, including the Florida Current, the Gulf of Mexico Loop Current, inshore currents of the SW Florida Shelf, discharge from the Everglades through the Shark River Slough, as well as by tidal exchange with both Florida Bay and Biscayne Bay [20,21,22].

In the present study, abiotic measurements were combined with microbial community analyses to analyze water quality in waters around the Florida Reef Tract.

## 2. Materials and Methods

### 2.1. Collection of Samples and Physico-Chemical Data

Water samples for this study (total of 50), were obtained between February and April of 2018. Water samples for microbial community analyses were collected at 30 stations from approximately 0.25 m below the surface and, if the stations were deep enough, also at approximately 1 m from the bottom, using a Niskin bottle (General Oceanics, Miami, FL, USA). The stations encompassed shore, inshore, and reef locations (Figure 1 and Appendix A). Water samples were collected and abiotic data were analyzed by the Southeast Environmental Research Center at Florida International University (SERC), using standard methodology outlined in the Quality Assurance Project Plan (QAPP, [22]). R package ‘vegan’ [23] was used to fit environmental variables (envfit function) onto ordinations of a detrended correspondence analysis (DCA) in order to determine their impact on bacterial and eukaryotic community composition.

### 2.2. Flow Cytometry

Flow cytometry samples were fixed with paraformaldehyde (final concentration 1%; pH = 7.4), incubated at room temperature (RT) for 60 min, and stored at −20 °C until analyses. Flow cytometry analyses were performed on a Guava easyCyte HT (Luminex, Austin, TX, USA). Samples for flow cytometry were incubated with SYBR Green I nucleic acid stain for 30 min at room temperature. Cell populations were discriminated via green fluorescence (532 nm), side scatter, and forward scatter channels using a blue laser (488 nm) at a flow rate of 0.24 μL s^−1^. Distinct microbial clusters, including low nucleic acid (LNA) and high nucleic acid (HNA) fractions that are often encountered in aquatic samples [24], were analyzed using Guava’s InCyte software (Luminex, Austin, TX, USA).

### 2.3. Phylogenetic Profiling

For the microbial community analyzes, 0.5 L of each sample was filtered onto 0.22 µm nitrocellulose membranes (MF-Millipore, Darmstadt, Germany) and stored at −20 °C until further processing. DNA extractions were carried out with the Qiagen PowerWater Kit following the manufacturer’s recommended protocol (Qiagen, Hilden, Germany).

DNA extracts were used as templates for the amplification of the V4 hypervariable region of the 16S rRNA gene (515F-806R primer pair [25]) and V9 hypervariable region of the 18S rRNA gene (1389F-EukB primer pair [26]). In addition, primers contained sequencer adapters and the reverse amplification primer contained a twelve base barcode sequence for multiplexing. Each 25 µL PCR reaction contained 9.5 µL of MO BIO PCR Water (Certified DNA-Free), 12.5 µL of QuantaBio’s AccuStart II PCR ToughMix (2x concentration, 1x final), 1 µL Forward Primer (5 µM concentration, 200 pM final), 1 µL Golay barcode tagged Reverse Primer (5 µM concentration, 200 pM final), and 1 µL of template DNA. The PCR conditions to amplify the 16S rRNA gene were as follows: 94 °C for 3 min, 35 cycles at 94 °C for 45 s, 50 °C for 60 s, and 72 °C for 90 s; final extension of 10 min at 72 °C to ensure complete amplification.

The PCR conditions to amplify the 18S rRNA gene were as follows: 94 °C for 3 min, 35 cycles at 94 °C for 45 s, 57 °C for 60 s, and 72 °C for 90 s; final extension of 10 min at 72 °C. Amplicons were then quantified using PicoGreen (Invitrogen, Carlsbad, CA, USA) and a plate reader (Infinite 200 PRO, Tecan, Männedorf, Switzerland). Once quantified, volumes of each of the products were pooled into a single tube so that each amplicon is represented in equimolar amounts. This pooled sample was then cleaned up using AMPure XP Beads (Beckman Coulter, Chaska, MN, USA), and quantified using a fluorometer (Qubit, Invitrogen, Carsbad, CA, USA). After quantification, the molarity of the pool was determined and diluted to 2 nM, denatured, and then diluted to a final concentration of 6.75 pM with a 10% PhiX spike. DNA sequence data were generated using Illumina paired-end sequencing (151 bp ⋅ 12 bp ⋅ 151 bp MiSeq run) at the Environmental Sample Preparation and Sequencing Facility at Argonne National Laboratory.

### 2.4. Bioinformatics

The QIIME 2 microbiome analysis package [27] was used for sequence analyzes. Quality filtering, chimera identification and merging of paired-end reads were carried out using the DADA2 plugin [28] as implemented in QIIME2 with default settings, except forward and reverse reads were truncated to 150 bp. The ‘core-metrics-phylogenetic’ method was used to obtain weighted UniFrac matrices with normalized sampling depths of 23,615 sequences in the case of 16S rRNA dataset and 39,703 sequences in the case of 18S rRNA dataset. SILVA release 132 (Ref NR 99) taxonomy and q2-feature-classifier were used for classification of gene sequences [29,30].

Data filtering and statistical analyses were carried out with R version 3.4.0 (R Core Team). Taxa classified as chloroplasts, mitochondria or Metazoa were discarded from the datasets. The final 16S rRNA dataset included 2,476,919 sequences, on average 49,538 sequences per sample. No sequences were retrieved from sample 224B. The final 18S rRNA dataset included 1,977,481 sequences, on average 39,549 sequences per sample. Raw data were submitted to the Sequence Read Archive under accessions SRR14089946-SRR14089995 (16S rRNA gene dataset) and SRR14090638-SRR14090687 (18S rRNA gene dataset). Dendrograms were constructed using unweighted pair group method with arithmetic mean (UPGMA) hierarchical clustering based on weighted UniFrac distance analyses. The approximately unbiased (AU) values and probability values (*p*-values) were calculated by multiscale bootstrap resampling. Clusters with AU ≥ 95% are considered to be strongly supported by data.

## 3. Results

### 3.1. Physical and Chemical Parameters

Sampling occurred at the end of the dry season at 30 stations along the Florida Keys archipelago. The locations were divided into four areas (Marquesas, Lower Keys, Middle Keys, and Upper Keys) and five zones (shore, inshore, reef, Marquesas Keys and the (Lower Keys) backcountry; Figure 1). Two shore stations were located at the Calusa Park Marina (Key Largo; station 501) and near the Key West International Airport (Key West; station 509). These two locations are arguably under the highest anthropogenic pressures among the study sites. The salinity ranged from 34.0 to 36.8 PSU (practical salinity units), and values lower than 36 PSU were observed only in five stations with shore stations among them (Figure 2 and Appendix A).

The Environmental Protection Agency (EPA) has developed strategic targets for the Water Quality Monitoring Project that state that they shall annually maintain the overall water quality of the near shore and coastal waters of the FKNMS according to the 2005 baseline. The baseline for dissolved inorganic nitrogen (DIN) has been set to 0.75 μM. Values exceeding this baseline were observed at one station in the Upper Keys (reef station 215) and at two stations in the Lower Keys (stations 301, 316; stations 501, 503 and 509 are not considered for the EPA targets). Higher total phosphorus (TP) concentrations than the 0.25 μM baseline were observed at the Calusa Park Marina (station 501; not considered for EPA targets) and at 10 stations in the Lower Keys. For reef sites, chlorophyll a (Chl a) values exceeding the baseline of 0.35 μg L^−1^ were observed at three reef stations in the Lower Keys (255, 256, and 280).

The water temperature at the stations varied by 4 degrees between 21.9 °C and 25.9 °C. Samples collected at shore and Lower Keys backcountry stations differed significantly by their physicochemical properties, including higher TOC content and increased turbidity (Figure 2). The highest Chl *a* and ammonium concentrations were also found among these stations (stations 295 and 509).

### 3.2. Abundance of Pico- and Nanoplankton

The heterotrophic, non-pigmented, bacterial cell abundances (picoplankton) were discriminated into low nucleic acid content (LNA) bacteria and high nucleic acid content (HNA) bacteria. For LNA bacteria, a subdivision was created to discriminate a distinct cluster containing notably larger cells with higher forward scatter values (LNA-hf; Figure 3). 

The total bacterial abundances ranged from 0.6 × 10^6^ to 1.2 × 10^6^ cells mL^−1^. The fraction of HNA bacteria varied greatly between 23% and 74% of the total bacterial counts. Highest bacterial cell counts were observed at inshore stations in the Upper and Middle Keys (214, 223, and 241), and at four stations in the Lower Keys backcountry (301, 305, 307, and 315). Lowest bacterial counts were found at reef station 280 (Eastern Dry Rocks) and at station 295 (Florida Bay).

Phytoplankton cells were discriminated by their size into pico- and nanoplankton groups (0.2–2.0 µm and 2.0–20 µm, Chla-nano). Picoplankton cells were also subdivided by their pigmentation: only Chl *a* containing cells (Chla-pico) and those additionally possessing phycoerythrin (Chla-pico+PE; Figure 3). Chla-pico was the most abundant phytoplankton group with cell densities between 3.6 × 10^3^ and 6.4 × 10^4^ cells mL^−1^. Chla-pico+PE abundances varied significantly, reaching 3.3 × 10^4^ cells at inner reef station 278, where it was the most abundant phytoplankton group. Highest nanophytoplankton (Chla-nano) abundances were observed at Lower Keys backcountry sites (up to 5.6 × 10^3^ cells mL^−1^), whereas phytoplankton abundances overall varied drastically between the stations (Figure 3).

### 3.3. Prokaryotic Community Composition (PCC)

In general, microbial communities (both prokaryotic and eukaryotic) did not form many significant clusters based on location or habitat (Figure 4), with few exceptions. PCC from all three stations in the Marquesas Keys and seven outermost reef stations from the Upper and Lower Keys formed a statistically significant cluster (edge 42; AU = 98%, Figure 4A), and shore station 501, inshore station 254, and four Lower Keys backcountry stations (301, 305, 307, and 316) formed another statistically significant cluster (edge 31; AU = 99%). The second shore station, 509, also clustered closely with this group (edge 37). Temperature (r^2^ = 0.55; *p* < 0.05), DO (r^2^ = 0.62; *p* < 0.01) and TOC (r^2^ = 0.78; *p* < 0.005) concentrations were physicochemical parameters that had a significant effect on PCC (Appendix A). Most surface and bottom layer samples collected from the same station clustered together, indicating that the waters were mixed well.

While the 16S rRNA gene amplicon analyzes revealed bacteria-dominated communities, Marine Group II archaea (*Thermoplasmata*) were present and had a relative abundance of 2.6% in the Upper Keys (and an average of 1.9% within the whole data set). The highest relative abundance of archaea was observed in three northernmost stations in the Upper Keys, where Marine Group II contributed to an average of 4.7% of the prokaryotic community composition (Figure 5).

Two bacterial classes, alpha- and gammaproteobacteria, together comprised 67.6% of the analyzed prokaryotic communities (Figure 5). The SAR11 clade (14.1%), *Rhodobacterales* (9.7%), *Puniceispirillales* (8.9%), *Rhodospirillales* (6.6%), and *Parvibaculales* (1.4%) were the most abundant orders within the alphaproteobacteria, while the SAR86 clade (12.5%), *Thiomicrospirales* (5.0%), *Oceanospirillales* (3.8%) and *Cellvibrionales* (2.4%) were the most abundant orders of Gammaproteobacteria. Together with *Flavobacteriales* (*Bacteroidia*; 14.1%), *Actinomarinales* (*Actinobacteria*; 5.9%), and *Synechococcales* (*Oxyphotobacteria*; 5.2%), these orders accounted for 87.9% of the prokaryotic diversity.

In the Upper Keys, the SAR11 clade (on average 15.7% of PCC), the SAR86 clade (11.6%), *Flavobacteriales* (11.1%), *Puniceispirillales* (9.7%), and *Rhodobacterales* (8.3%) were the most prevalent bacterial orders. *Synechococcales* accounted for a larger fraction of the PCC (on average 6.5%) in the Upper Keys compared to other study areas, with a maximum occurrence of 14% at reef station 222.

In the Middle Keys, *Thiomicrospirales* was the most abundant bacterial order, contributing 19.6% of the PCC (up to 31.4% at inshore station 244), whereas in other study areas, its members were significantly less abundant (below 4% of the PCC). Representatives of *Betaproteobacteriales* were most commonly found in this study area, accounting for 1.2% of the PCC, whereas their abundance in other areas remained less than 1%.

In the Lower Keys, members of *Flavobacteriales* represented around one-fifth of the PCC (20.1%), roughly two-fold higher compared to other study areas. *Rhodobacterales* (13.1%) was the second most abundant microbial order, and its relative abundance in the Lower Keys was around five-fold higher than in the Marquesas Keys (2.4%), where SAR86 and SAR11 clades were the dominant taxa (28.2% and 19.0%, respectively).

SAR11 bacteria were represented by seven relatively abundant species-level taxa belonging to four different clades (Ia, Ib, II, III and IV) and their occurrence varied between its ecotypes (Figure 5). Within the cyanobacteria, *Synechococcus* generally dominate in coastal waters and *Prochlorococcus* in offshore oligotrophic regions [31]. This trend is also reflected in our results as the highest relative abundances of *Prochlorococcus* were observed at outer reef stations (203, 206, 222, 243, 256, and 280).

### 3.4. Eukaryotic Microbial Community (EMC)

Eukaryotic microbial communities showed little clustering with highly significant statistical support, neither based on location nor on habitat (Figure 4B). The largest statistically significant cluster (edge 27; AU = 98%) was comprised of the four samples from stations 242 and 244 (Middle Keys). Similarly to the PCC, TOC (r^2^ = 0.65; *p* < 0.001), temperature (r^2^ = 0.46; *p* < 0.05) and DO (r^2^ = 0.55; *p* < 0.05) had the most significant impact on the EMC (Appendix A).

Dinoflagellates represented more than a quarter of the EMC, with *Dinophyceae* and *Syndiniales* comprising 13.8% and 13.0% of the 18S rRNA dataset, respectively (Figure 6). Dinoflagellates that remained unclassified accounted for an additional 2.1% of the EMC structure. The second most abundant phytoplankton group were diatoms. Members of Bacillariophyta accounted for 16.8% of the EMC composition. Other relatively abundant taxa, that contributed more than 1% of the dataset, were marine *Stramenopiles* (MAST group; 8.1%), *Mamiellophyceae* (*Chlorophyta*; 7.0%), *Spirotrichea* (*Ciliophora*; 5.5%), *Prymnesiophyceae* (*Haptophyta*; 3.5%), Fungi (2.1%), *Labyrinthulea* (*Stramenopiles*; 2.0%), *Chrysophyceae* (1.8%), *Telonemia* (1.3%), *Ascetosporea* (*Endomyxa* 1.3%), *Cryptophyceae* (1.3%), *Apicomplexa* (1.2%), and *Thecofilosea* (*Cercozoa*; 1.0%).

The most abundant genus-level taxa within the EMC varied vastly in their relative abundance throughout the study area, except for the most prominent dinoflagellate, *Scrippsiella* (OTU1), and a raphid-pennate diatom, OTU2, that were commonly found in all the samples (Figure 6). The most dominant taxon in a single location was OTU3, *Syndiniales* of marine alveolate group I clade I, that comprised about one-third (33.1%) of the EMC at the bottom layer of inner reef station 274.

Within *Bacillariophyta*, raphid-pennate diatoms were the most abundant group represented by OTU2 and OTU19 (Figure 6). OTU2 was relatively abundant in all study areas and contributed to around a quarter of the EMC at reef station 280. Other abundant OTUs related to diatoms included representatives of the genus *Minidiscus* (OTU4), which contains species that represent the smallest centric diatoms of the marine phytoplankton [32]. *Chaetoceros*, which is probably the most diverse genus of marine planktonic diatoms, was more prevalent at stations in Middle and Marquesas Keys than in the other locations (OTU14).

Several sequences were related to green algae: OTU5, OTU6, and OTU20 were classified as *Micromonas*; OTU33 was identified as *Ostreococcus*.

The most abundant sequences related to ciliates were assigned to the genus *Spirotrichea*, which includes mixotrophs that can use plastids from their prey as well as retain chloroplasts from food for photosynthetic nutritional supplements [33]. Another OTU assigned to ciliates was closely related to *Laboea strobila* (OTU34), an organism that sequesters photosynthetically functional chloroplasts derived from ingested algae [34].

## 4. Discussion

The waters around the Florida Reef Tract are generally oligotrophic and nutrient-deplete [22]. Year-long seasonal monitoring of abiotic water quality parameters (including the dataset presented here) has shown that waters have usually slightly higher turbidity and nutrient concentrations on the Gulf of Mexico side of the Keys than on the Atlantic side, along the reef tract [22]. Despite complex water patterns and significant differences in anthropogenic pressure along the Florida Reef Tract, the microbial community composition, at least during our sampling event, was not strikingly different in the different waters around the Keys and no distinct coherent clustering by longitude/latitude or sample type was apparent.

Elevated organic carbon concentrations can directly impact coral microbiomes and increase coral mortality [35]. The importance of increased organic carbon for microbial community structure in reef waters was well demonstrated by our data, as TOC concentrations did have a significant impact on both prokaryotic and eukaryotic microbial community composition, together with temperature and DO concentrations. 

The data presented in this study show that the total abundances of unpigmented cells are not necessarily correlated with organic carbon concentrations, and even though higher abundances of HNA bacteria were more likely to be found in areas with elevated TOC concentrations, this trend was not consistent. More detailed studies on microbial functions rather than relative abundances of certain microorganisms are necessary to address questions related to changes in functional diversity of planktonic microbes near healthy and unhealthy reefs.

Nevertheless, previous data on microbial community composition in reef environments have identified potential bioindicator species and their relationship to abiotic stressors. Existing monitoring data that combine microbial data and abiotic data in reef habitats demonstrated that high temperatures are usually correlated to an increase in taxa belonging to *Rhodobacteraceae*, *Cryomorphaceae*, *Synechococcaeae*, *Vibrio* and *Flavobacterium* (which include putative coral pathogens and opportunistic bacteria). *Flavobacteriaceae*-affiliated taxa are significant indicators correlated to high chlorophyll a (Chl a), TSS and particulate organic carbon (POC) concentrations. *Halomonadaceae* are significantly correlated with high Chl a and TSS, and representatives of the phylum *Verrucomicrobia* are significant indicators correlated with high TSS levels [19]. 

Opportunistic copiotrophic taxa, such as *Cryomorphaceae*, *Flavobacteriaceae* and *Rhodobacteraceae,* are usually more prevalent in the higher nutrient nearshore waters (e.g., [36,37]). Recent studies have pinpointed *Flavobacteriaceae*-affiliated taxa as indicators for increased organic nutrients at the Great Barrier Reef [19,38]. We found that the relative abundances of *Cryomorphaceae* PS008 were significantly correlated with TOC concentrations in the water (R^2^ = 0.77, *p* < 0.001). Interestingly, several bacterial groups within the order *Flavobacteriales*, including *Cryomorphaceae*, are more likely to be present within SCTLD-diseased coral tissue than within apparently healthy coral tissue [6]. Greater abundances of *Rhodobacteraceae* and *Cryomorphaceae* have been also shown at inlet-influenced coastal waters of southeast Florida, where the SCTLD outbreak began [39,40]. PS001, classified as *Rhodobacteraceae*, was the most abundant prokaryotic species-level taxa in our dataset and was found relatively abundant throughout the Florida Keys archipelago. *Rhodobacterales* have been also detected at higher relative abundances in SCTLD lesion samples [8]. Usually, bacteria associated with coral disease are rarely detected in seawater due to their low concentrations [41,42]. In this case, the same taxa that are abundant in corals with SCTLD symptoms are abundantly found in the water column, which further supports the hypothesis that some of these organisms may act as secondary opportunistic pathogens associated with progression of SCTLD [43].

Terrestrial runoff that leads to organic enrichment of coastal waters and sediments has been identified as a key process in the degradation of coral reefs [44,45]. Increases in organic matter result in reduced O_2_ concentrations, lower pH, and formation of hydrogen sulfide, a potent toxin to most organisms, which can accelerate the spread of reef colony mortality [44]. The high relative abundance of potentially sulfur-oxidizing *Thioglobaceae* (PS007) in the water column indicates high sulfide production in areas where SCTLD had spread at the time of the study (Upper and Middle Keys; [40]).

Not only prokaryotes, but also microbial eukaryotes are being used to assess water quality in coral reef ecosystems [46]. Microalgae are generally enriched nearshore due to high nutrient and resuspension requirements [47]. The most abundant eukaryotic group in our dataset, *Scrippsiella*, is a non-toxic, cosmopolitan marine dinoflagellate that can be found in both cold and tropical waters, where it is known to produce “red tide” events. *Scrippsiella* blooms can lead to oxygen depletion, resulting in fish kills [48], and have been reported in the Southern Gulf of Mexico and the coastal United States [49].

Dinoflagellates of the order *Suessiales*, which also contains the genus *Symbiodinium*, the main phototrophic coral symbiont, were also found in the water column. High nutrient concentrations [50,51,52,53], but also high numbers of dinoflagellates in reef waters impose potential threats to *Symbiodinium*. Firstly, it may lead to the spread and proliferation of viruses that could also infect zooxanthellae species [54,55,56], and secondly, it may increase activity and impact of algicidal bacteria against dinoflagellates [57]. While Mayali and Azam [58] initially reported that species belonging to the *Bacteroides* are the most abundant and widely isolated algicidal bacteria, a recent study identified culturable algicidal bacteria from a wide range of taxa [59].

Dinoflagellates are known as one of the major components of diverse marine ecosystems (e.g., [60]). They often form red tides or harmful algal blooms that sometimes cause human illness and large-scale mortality of fin-fish and shellfish [61,62,63]. Thus, the high abundance of dinoflagellates in the waters of a region that has a large tourism industry is of critical concern, not only in relation to SCTLD, but also in relation to other human interests. Our study indicates that dinoflagellates contributed to a larger fraction of the EMC compared to abundances reported by a study that was carried out in the Florida Keys three years earlier [64], although this could reflect seasonal variation or differences in the composition of sampling locations. 

Several abundant eukaryotic OTUs were assigned to well-described pathogens. Four OTUs were classified as *Syndiniales*, an order of dinoflagellates exclusively composed of marine parasites that infect a wide range of hosts, from fish larvae to dinoflagellates, including *Scrippsiella* [65,66]. Other parasitic eukaryotes in our dataset included *Paradinium* sp. (parasites of various copepod hosts, [67]), *Rozella* (*Cryptomycota*), a genus of endoparasites of a broad range of hosts [68], including other parasites [69], and Labyrinthulids, (endophytic net slime molds, most of which are opportunistic pathogens found in association with marine vegetation, including seagrasses and mangroves [70,71]). As our dataset only provides a snapshot of the microbial communities in these waters, establishing a regular monitoring program would be essential to assess changes in the relative abundances of these organisms.

## 5. Conclusions

A growing body of research has introduced the concept of using microorganisms as bioindicators, which can provide an immediate and sensitive measure of water quality that can and should supplement abiotic water quality measurements. Waters surrounding reefs interact with different components of coral reefs and thus may have a strong impact on reef health. Our survey of the waters around the Florida Keys uncovered a high abundance of copiotrophic microbial taxa, including opportunistic pathogens. This survey represents only a snapshot of the potential factors in the waters that might influence reef health in the Florida Keys’ archipelago, and regular monitoring of microbes in conjunction with abiotic stressors will be pivotal to understand possible threats to reef health and can thus guide informed ecosystem management.

## Figures and Tables

**Figure 1 microorganisms-09-01120-f001:**
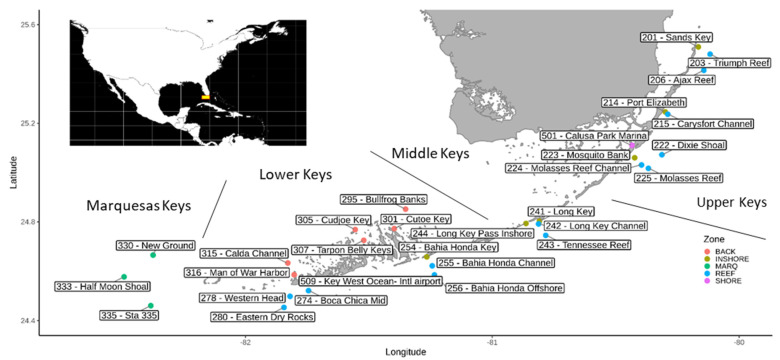
Map of the sampling sites (main graph) and location of the Florida Keys (upper left corner). Sampling sites are color-coded: BACK—Backcountry; INSHORE—Inshore; MARQ—Marquesas Keys; REEF—Reef; SHORE—Shore.

**Figure 2 microorganisms-09-01120-f002:**
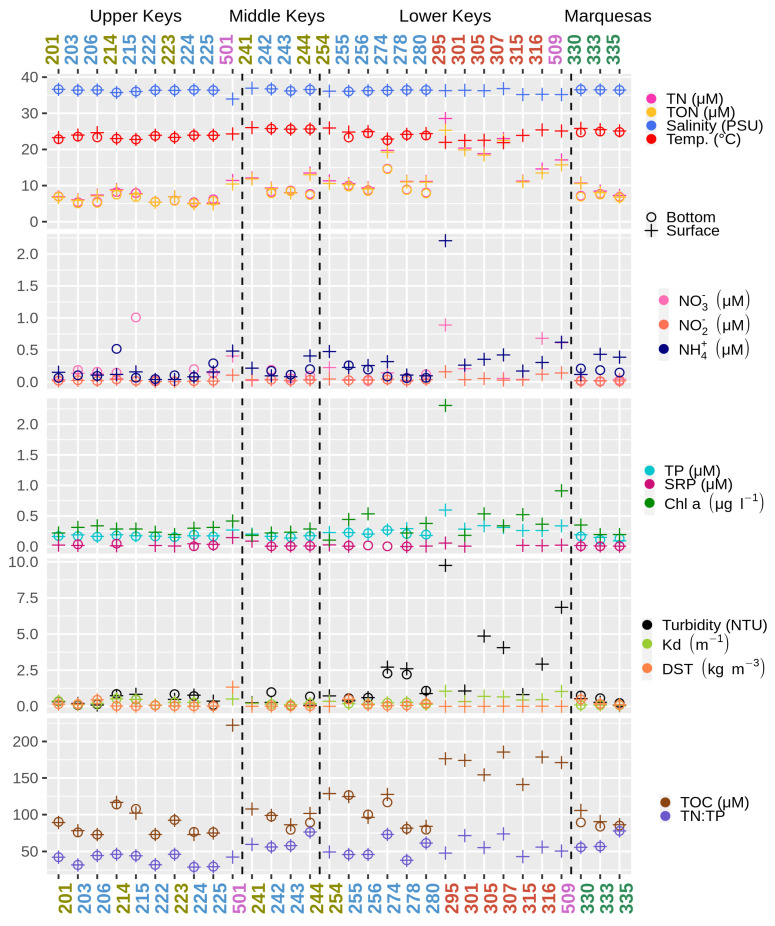
Physicochemical parameters of the water samples: total nitrogen (TN), total organic nitrogen (TON), nitrate, nitrite, ammonium, total phosphorus (TP), soluble reactive phosphorus (SRP), chlorophyll a (Chl a), diffuse attenuation coefficient (Kd), Delta Sigma-T (DST), total organic carbon (TOC) and total nitrogen to total phosphorus ratio (TN:TP). The color code for the stations follows Figure 1.

**Figure 3 microorganisms-09-01120-f003:**
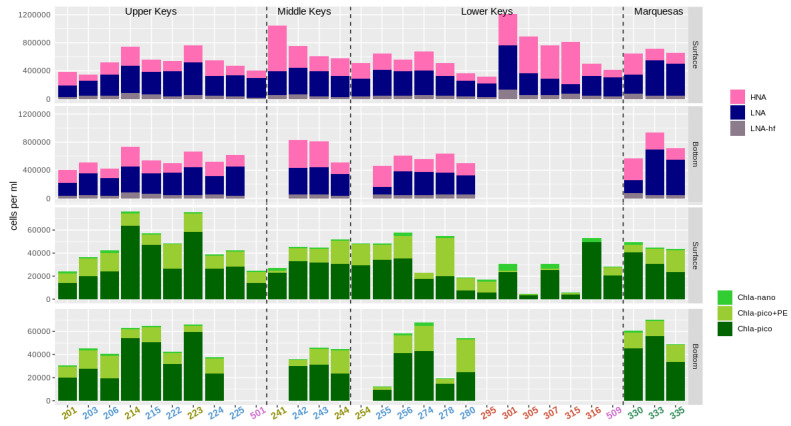
Flow cytometric estimates of cell abundances. Non-pigmented prokaryotes are divided between low nucleic acid (LNA) and high nucleic acid (HNA) fractions. Autofluorescent phytoplankton cells are discriminated into three groups: chlorophyll a containing picoplankton (Chla small), chlorophyll a containing nano- and microplankton (Chla big) and phycoerythrin containing pico- to microplankton (Chla+PE). The color coding for the stations follows Figure 1. Lower Keys sites and site 501 were only sampled at the surface. The analysis of unstained samples from site 225 did not work because of a failure of the flow cytometer.

**Figure 4 microorganisms-09-01120-f004:**
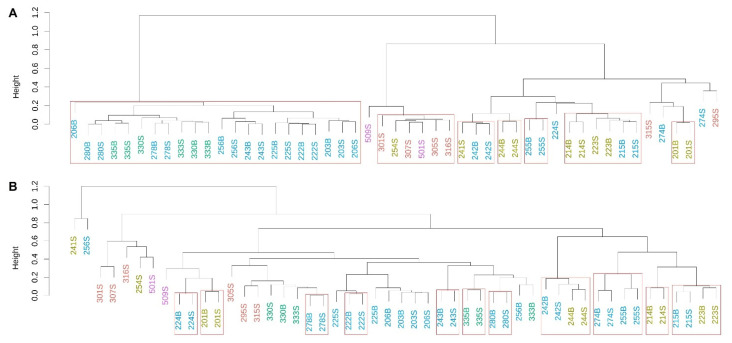
Cluster dendrograms (unweighted pair group method with arithmetic mean, UPGMA) based on weighted UniFrac distance matrices for prokaryotic (**A**) and eukaryotic (**B**) communities. Significant clusters (approximately unbiased *p*-values, AU, ≥95%, 1000 iterations) are indicated with by red boxes. The color code for the stations follows Figure 1.

**Figure 5 microorganisms-09-01120-f005:**
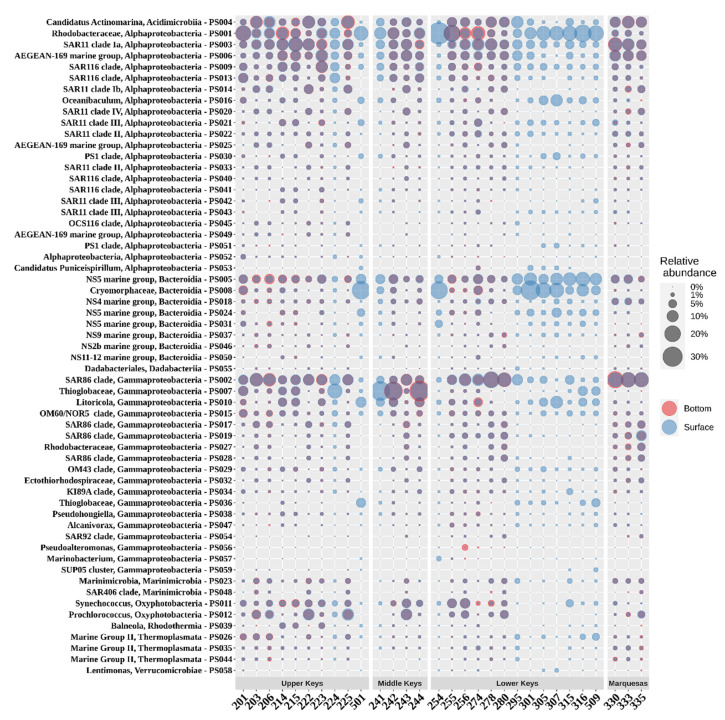
Relative abundance of the most abundant prokaryotic taxa at species level (accounted for at least 1% in one sample).

**Figure 6 microorganisms-09-01120-f006:**
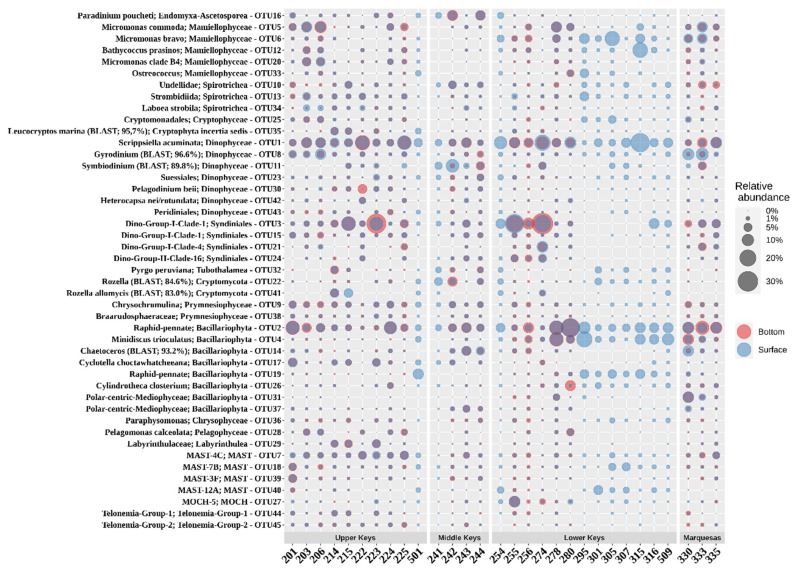
Relative abundance of eukaryotes (genus-level Operational Taxonomic Units, OTUs, clustered at 95% identity) that contributed at least 5% of community composition in at least one sample.

## Data Availability

Raw data was submitted to the Sequence Read Archive under accessions SRR14089946-SRR14089995 (16S rRNA dataset) and SRR14090638-SRR14090687 (18S rRNA dataset).

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
