# Peer review of "Composition of Prokaryotic and Eukaryotic Microbial Communities in Waters around the Florida Reef Tract"

_microorganisms, 2021, doi:10.3390/microorganisms9061120_

Round 1

Reviewer 1 Report

This paper is an interesting topic that is definitely relevant for monitoring and management of the coral reefs in the FKNMS and nearby waters. I think it is important to understand the environmental and anthropogenic causes behind the onset of SCTLD as well as the changes in the 'coral ecosphere', to assist in possible mitigation. I was left wondering just a few minor things after reading this paper. The mention of seasonality in the discussion was quite important and I was wondering why the authors chose to only sample February to April if seasonality could be a contributing factor to SCTLD? If there was a reason behind the period of sampling, perhaps due to certain environmental occurrences during this time, such as Red Tide, etc., there was no real explanation behind this. Perhaps in the future, the authors should consider a more well-rounded seasonal sampling approach to really understand the changes occurring in the FKNMS, in relation to the onset of SCTLD. I feel this is quite an important issue that should be further explored. Lastly, I am not sure if it was the quality of the pdf I received, however, all figures in this MS were incredibly difficult to read, with the exception of the supplemental figures, and the low quality made it hard to explore the data further. 

Author Response

This paper is an interesting topic that is definitely relevant for monitoring and management of the coral reefs in the FKNMS and nearby waters. I think it is important to understand the environmental and anthropogenic causes behind the onset of SCTLD as well as the changes in the 'coral ecosphere', to assist in possible mitigation. I was left wondering just a few minor things after reading this paper. The mention of seasonality in the discussion was quite important and I was wondering why the authors chose to only sample February to April if seasonality could be a contributing factor to SCTLD?

We want to thank this reviewer for the comments. As all other reviewers mentioned the lack of data related to SCTLD, we refocused and shortened the text. We simply do not have associated data on SCTLD and therefore the critique on linking our data to SCTLD is well justified. The sampling dates were aligned with ongoing, quarterly (abiotic) water quality studies conducted by FIU.

 If there was a reason behind the period of sampling, perhaps due to certain environmental occurrences during this time, such as Red Tide, etc., there was no real explanation behind this. Perhaps in the future, the authors should consider a more well-rounded seasonal sampling approach to really understand the changes occurring in the FKNMS, in relation to the onset of SCTLD. I feel this is quite an important issue that should be further explored.

Thanks for this comment, we would love to do that in the future. Also, more analyses of waters on/above corals as well as associated studies on coral health would certainly be necessary. We see this study as an important snapshot of microbial communities in these waters, and hopefully be able to establish better datasets in the near future, together with local collaborators.

Lastly, I am not sure if it was the quality of the pdf I received, however, all figures in this MS were incredibly difficult to read, with the exception of the supplemental figures, and the low quality made it hard to explore the data further. 

We apologize; another reviewer commented on this issue as well. This must have gone wrong during the editing process as the figures in the preprint, which used exactly the same word document, are of good quality. We embedded the figures again, and uploaded them as well. We hope this improves the quality.

Reviewer 2 Report

In this study, 30 water samples from different stations along the FL keys were collected to conduct microbial and physicochemical analysis to understand correlations between the two. However, this study sets itself up to be about SCTLD and the coral ecosphere but the analysis, results, and methods have no relation to these topics. If the authors would like to understand SCLTD in relation to their sampling scheme they could identify the stage of disease progression of SCTLD in the Fl Keys during the collections of their samples and try to correlate the microbiome based on these different stages (i.e., epidemic, endemic, vulnerable) similar to the methods used in https://doi.org/10.3389/fmicb.2020.00681, but adding the physicochemical information. In addition, the authors focus a lot of time on the coral ecosphere, but from my understanding, none of the water samples were sampled right above a coral colony and thus have nothing to do with the coral ecosphere. I think there is potential to get interesting results from this dataset, but the entire manuscript is disconnected and needs a cohesive direction. The discussion also reads like a review paper and there are is little discussion about the results of the paper. 

Author Response

In this study, 30 water samples from different stations along the FL keys were collected to conduct microbial and physicochemical analysis to understand correlations between the two. However, this study sets itself up to be about SCTLD and the coral ecosphere but the analysis, results, and methods have no relation to these topics. If the authors would like to understand SCLTD in relation to their sampling scheme they could identify the stage of disease progression of SCTLD in the Fl Keys during the collections of their samples and try to correlate the microbiome based on these different stages (i.e., epidemic, endemic, vulnerable) similar to the methods used in https://doi.org/10.3389/fmicb.2020.00681, but adding the physicochemical information. In addition, the authors focus a lot of time on the coral ecosphere, but from my understanding, none of the water samples were sampled right above a coral colony and thus have nothing to do with the coral ecosphere. I think there is potential to get interesting results from this dataset, but the entire manuscript is disconnected and needs a cohesive direction. The discussion also reads like a review paper and there are is little discussion about the results of the paper.

We thank this reviewer for the comments. In line with comments from most other reviewers, we refocused tile and text towards general water quality and microbial community composition and away from SCTLD. While we certainly don’t argue the fact that we do not present any data on SCTLD or even coral health, we still strongly believe that the waters around the reefs have an impact on reef health (and possible SCTLD). We also strongly believe that SCTLD is the most important environmental issue in the Florida Reef Tract right now, and thus any paper discussing water quality issues in this environment should put those into context with SCTLD. Nevertheless, we shortened and re-focused the manuscript accordingly.

Reviewer 3 Report

I found this study to be a great assessment of the water quality of the Florida Keys, however, I am deeply troubled by the forced connections to SCTLD made in this paper. This is mostly because there is simply no data collected on SCTLD, so how can any connections be made to the data collected and the ongoing disease outbreak. Yes, data was collected across the Keys over a three-month period, but what was going on with the outbreak then? I found it inappropriate to used SCTLD in the title, introduction, and so heavily in the discussion to bring attention to this study without gathering any data on SCTLD. I can see this manuscript published as an assessment of water quality with a brief mention of SCTLD.

Here are my specific comments:

  • Line 28: Is there a citation for this?
  • For SCTLD, the words in the name for this disease are not capitalized according to the guidelines put forth by management agencies overseeing work on this outbreak.  
  • Line 49: "microbial origin" - specifically bacteria in origin. Please change to be more accurate.
  • Line 51-56: This seems out of place since these stressors have yet to be linked to SCTLD. Therefore, I find it a bit misleading and unnecessary to have this paragraph.
  • Line 59: "pathogen-like microorganism" is not a used term. Please use something like "potentially pathogenic" or "...potentially increases the prevalence of copiotrophic pathogenic microorganisms. "
  • Materials and Methods: The most worrisome thing I noticed about the methods is there is a complete lack of any data collection on SCTLD. Not even surveys on occurrence or prevalence. How can this data be even connected to SCTLD? There is no data connecting any of the findings in this paper to SCTLD. This is an assessment of the water quality of the FKNMS, not a study on SCTLD.
  • Starting at line 83 - collection methods: I'm immediately concerned that wide-sweeping claims are being made about a disease, location, and ecosystem-based on a 3-month sampling period. Even more concerning is that it is immediately not clear if there were baseline or pre-SCTLD samples taken. Furthermore, this outbreak has been occurring since 2014, how can a clear assessment of "transmission and progression of SCTLD" with such a relativity short temporal period?
  • Line 317: This is simply not true. This disease was found to be transmissible by various studies.
  •  Throughout this discussion, I could not shake the fact that I saw absolutely no actual data collected on SCTLD and I don't agree with including the claims that any of this data has a connection to the disease. How can you make an assessment of the health of the Keys if there are no baseline measurements? Any data on the status of SCTLD at sites during collections? What is the overall health or relative health of the reefs during sampling? Do these results even correlate with reef health?

Author Response

I found this study to be a great assessment of the water quality of the Florida Keys, however, I am deeply troubled by the forced connections to SCTLD made in this paper. This is mostly because there is simply no data collected on SCTLD, so how can any connections be made to the data collected and the ongoing disease outbreak. Yes, data was collected across the Keys over a three-month period, but what was going on with the outbreak then? I found it inappropriate to used SCTLD in the title, introduction, and so heavily in the discussion to bring attention to this study without gathering any data on SCTLD. I can see this manuscript published as an assessment of water quality with a brief mention of SCTLD.

We thank this reviewer for the comments. In line with comments from other reviewers, we refocused tile and text towards general water quality and microbial community composition and away from SCTLD. While we certainly don’t argue the fact that we do not present any data on SCTLD or even coral health, we still strongly believe that the waters around the reefs have an impact on reef health (and possible SCTLD). We also strongly believe that SCTLD is the most important environmental issue in the Florida Reef Tract right now, and thus any paper discussing water quality issues in this environment should put those into context with SCTLD. Nevertheless, we shortened and re-focused the manuscript accordingly.

Here are my specific comments:

Line 28: Is there a citation for this?

Added Finkl and Andrews 2008 as reference.

For SCTLD, the words in the name for this disease are not capitalized according to the guidelines put forth by management agencies overseeing work on this outbreak. 

Thank you. Changed in the text.

Line 49: "microbial origin" - specifically bacteria in origin. Please change to be more accurate.

Changed.

Line 51-56: This seems out of place since these stressors have yet to be linked to SCTLD. Therefore, I find it a bit misleading and unnecessary to have this paragraph.

As we re-focused the text on water quality issues rather than SCTLD, we think this paragraph is still valuable. We modified the text in lines 50-64 to make it clear that these factors do not seem to be linked to SCTLD.

Line 59: "pathogen-like microorganism" is not a used term. Please use something like "potentially pathogenic" or "...potentially increases the prevalence of copiotrophic pathogenic microorganisms. "

Changed accordingly.

Materials and Methods: The most worrisome thing I noticed about the methods is there is a complete lack of any data collection on SCTLD. Not even surveys on occurrence or prevalence. How can this data be even connected to SCTLD? There is no data connecting any of the findings in this paper to SCTLD. This is an assessment of the water quality of the FKNMS, not a study on SCTLD.

We completely agreed, it is an assessment of the water quality of the FKNMS. We removed the focus on SCTLD in title and text. See general comment above.

Starting at line 83 - collection methods: I'm immediately concerned that wide-sweeping claims are being made about a disease, location, and ecosystem-based on a 3-month sampling period. Even more concerning is that it is immediately not clear if there were baseline or pre-SCTLD samples taken. Furthermore, this outbreak has been occurring since 2014, how can a clear assessment of "transmission and progression of SCTLD" with such a relativity short temporal period?

We assessed microbial community patterns in these waters, and highlighted potential organisms that might have an impact on this disease. But, in line with this comment and comments of other reviewers, we re-focused the text, changed the title and shortened the references related to SCTLD.

Line 317: This is simply not true. This disease was found to be transmissible by various studies.

We deleted this paragraph.

Throughout this discussion, I could not shake the fact that I saw absolutely no actual data collected on SCTLD and I don't agree with including the claims that any of this data has a connection to the disease. How can you make an assessment of the health of the Keys if there are no baseline measurements? Any data on the status of SCTLD at sites during collections? What is the overall health or relative health of the reefs during sampling? Do these results even correlate with reef health?

The short answer to this is: We don’t know and thus we shortened the text related to SCTLD.

Reviewer 4 Report

The title and framing of this article (to link it to stony coral tissue loss disease; SCTLD) is not supported by the data and results presented. There is very little to link these data to SCTLD and while that can still be mentioned via a sentence or two in the discussion, it is purely speculative and therefore should not be the focus of the paper.

The data provide a geographic snapshot of microbial community composition and water quality parameters for the Florida Reef Tract in 2018. Recommend the title be limited to that. This still fits well with the special topic issue towards which the paper was submitted.

The abstract (lines 15-18), introduction (lines 70-72), and discussion (lines 343, 387) repeatedly bring up the ‘coral ecosphere,’and appear to apply this concept to the data in the paper. However, none of the samples in this study qualify as being from the coral ecosphere. The term (as used in Weber et al. 2019, DOI 10.1002/lno.11190) refers to a boundary layer surrounding individual coral colonies, roughly within 30 cm of the coral surface. Note that earlier work attempting to investigate coral influence on reef water (e.g., Seymour et al. 2005, MEPS Vol. 288: 1–8, 2005;  Tout et al. 2014, DOI 10.1007/s00248-013-0362-5) collected water samples within 1-12 cm of the coral surface. These studies compared those ‘coral-influenced’ samples against reef water samples collected from 1 meter away. All of the water samples discussed in this manuscript are either from just below the surface or from 1 meter above bottom; these are reef water samples, not coral ecosphere.

In 2018, SCTLD had burned through the Upper and Middle Keys but not most of the Lower Keys or Marquesas. This is barely mentioned in spite of the paper’s framing (likely because the collected data do not show any major differences between the areas; only mention is Discussion lines 373-375: “The presence of potentially sulfur-oxidizing Thioglobaceae (PS007) in the water column indicates high sulfide production in areas where SCTLD had spread at the time of the study (Upper and Middle Keys; [43].”

As such, suggest reframing the paper to be more like similar studies that have collected reef water samples and used microbial and water quality parameters to place the overall reef health in context (e.g., by comparing against older datasets, comparing against other reefs in the region, comparing Upper vs. Middle vs. Lower). Examples include Bruce et al. 2012 DOI 10.1371/journal.pone.0036687, Weber et al. 2019 DOI 10.1111/1462-2920.14870, Paul et al. 1995 AEM 61: 2235-2241). Hurricane Irma passed through the Florida Keys in September 2017; could that have impacted trends seen in these data (e.g., lines 180-183)?

Specific comments:

Figures (particularly 1, 4 and 6) are too low resolution to read key information (like the color legend). Fortunately, the authors published as a preprint and the figures in that document are crystal clear, even when enlarged on screen.

Introduction – Line 34, suggest that in addition to a 2005 reference, the authors include the latest effort: https://www.fisheries.noaa.gov/southeast/habitat-conservation/restoring-seven-iconic-reefs-mission-recover-coral-reefs-florida-keys

Line 40: Provide URL or more complete citation information for (NOAA, 2019)

Lines 53-57: Note that bleaching can influence subsequent coral disease mortality: Muller et al. 2008 DOI 10.1007/s00338-007-0310-2, Miller et al. 2009 DOI 10.1007/s00338-009-0531-7. Also that better references for influence of nutrients include: Vega Thurber et al. 2013 DOI 10.1111/gcb.12450, Zaneveld et al. 2016, DOI 10.1038/ncomms11833.

Lines 67-75: See comments above about the difference between ‘coral ecosphere’ and reef waters.

Line 73: What does reference 21 (the structure of bacterial communities in arctic ocean water) have to do with coral reefs or reef condition?

Lines 79-80: The data do not address transmission or progression of SCTLD

Methods – Lines 134-135 Insufficient information to allow someone to reproduce your bioinformatic analyses in QIIME2. See Reigel et al. 2020 DOI 10.1007/s00338-020-02006-5 for a good example.

Results

Table S1: Confirming that station depth data are in meters (unit not specified)?

Line 216: “from stations from all three stations”

Figure 4: People with red/green color blindness will not be able to distinguish between the two numbers (suggest you chose a different color combination and/or augment by specifying left and right). Also, consider highlighting the few significant clusters by putting circles around them or light color block behind them.

Line 239: Should reference Figure 5 not Figure 4?

Discussion

Lines 328-331: See Kline et al. 2006 MEPS 314: 119-125.

Lines 343-346: Reference? Also see Maher et al. 2020 DOI 10.3389/fevo.2020.555698

Lines 365-368: Highly speculative statement considering the shared taxa being discussed are at the order or family level (each of which contain lots of diversity at the genus, species, and strain level).

Lines 373-374: Interesting but speculative. If Thioglobaceae (PS007) is indicative of high sulfide, shouldn’t it correlate with low DO?

Conclusions – Line 416: You did not survey the coral ecosphere.

Author Response

The title and framing of this article (to link it to stony coral tissue loss disease; SCTLD) is not supported by the data and results presented. There is very little to link these data to SCTLD and while that can still be mentioned via a sentence or two in the discussion, it is purely speculative and therefore should not be the focus of the paper.

Agreed and corrected throughout the text.

The data provide a geographic snapshot of microbial community composition and water quality parameters for the Florida Reef Tract in 2018. Recommend the title be limited to that. This still fits well with the special topic issue towards which the paper was submitted.

Changed the title accordingly.

The abstract (lines 15-18), introduction (lines 70-72), and discussion (lines 343, 387) repeatedly bring up the ‘coral ecosphere,’and appear to apply this concept to the data in the paper. However, none of the samples in this study qualify as being from the coral ecosphere. The term (as used in Weber et al. 2019, DOI 10.1002/lno.11190) refers to a boundary layer surrounding individual coral colonies, roughly within 30 cm of the coral surface. Note that earlier work attempting to investigate coral influence on reef water (e.g., Seymour et al. 2005, MEPS Vol. 288: 1–8, 2005;  Tout et al. 2014, DOI 10.1007/s00248-013-0362-5) collected water samples within 1-12 cm of the coral surface. These studies compared those ‘coral-influenced’ samples against reef water samples collected from 1 meter away. All of the water samples discussed in this manuscript are either from just below the surface or from 1 meter above bottom; these are reef water samples, not coral ecosphere.

We apologize for this, and use ‘reef waters’ now  throughout the manuscript to refer to our samples. We also added references that demonstrate the impact of the (abiotic and biotic) water quality around reefs for reef health (Lesser 2006; Garren and Azam 2012, Glasl et al 2017). We deleted the sections on coral ‘ecosphere’.

In 2018, SCTLD had burned through the Upper and Middle Keys but not most of the Lower Keys or Marquesas. This is barely mentioned in spite of the paper’s framing (likely because the collected data do not show any major differences between the areas; only mention is Discussion lines 373-375: “The presence of potentially sulfur-oxidizing Thioglobaceae (PS007) in the water column indicates high sulfide production in areas where SCTLD had spread at the time of the study (Upper and Middle Keys; [43].”

As such, suggest reframing the paper to be more like similar studies that have collected reef water samples and used microbial and water quality parameters to place the overall reef health in context (e.g., by comparing against older datasets, comparing against other reefs in the region, comparing Upper vs. Middle vs. Lower). Examples include Bruce et al. 2012 DOI 10.1371/journal.pone.0036687, Weber et al. 2019 DOI 10.1111/1462-2920.14870, Paul et al. 1995 AEM 61: 2235-2241).

Agreed, re-focused accordingly.

Hurricane Irma passed through the Florida Keys in September 2017; could that have impacted trends seen in these data (e.g., lines 180-183)?

Thanks for this interesting remark. As we don’t have any associated data on Hurricane Irma either, we do not want to speculate further. There is a growing body of literature that shows that storm events can mobilize legacy nutrients from the sediments, which leads to short- and mid-term changes in microbial community structure, but we feel that further speculation will not be beneficial for this manuscript.

Specific comments:

Figures (particularly 1, 4 and 6) are too low resolution to read key information (like the color legend). Fortunately, the authors published as a preprint and the figures in that document are crystal clear, even when enlarged on screen.

The same pdf was used for the preprint, so most likely there was an issue during editing. We re-embedded the figures, uploaded them separately as well, and informed the editor.

Introduction – Line 34, suggest that in addition to a 2005 reference, the authors include the latest effort: https://www.fisheries.noaa.gov/southeast/habitat-conservation/restoring-seven-iconic-reefs-mission-recover-coral-reefs-florida-keys

These efforts are fantastic and we are very happy to hear that this is happening now. We included this webpage now as a reference.

Line 40: Provide URL or more complete citation information for (NOAA, 2019)

We refer to a report now on lines 33-34.

Lines 53-57: Note that bleaching can influence subsequent coral disease mortality: Muller et al. 2008 DOI 10.1007/s00338-007-0310-2, Miller et al. 2009 DOI 10.1007/s00338-009-0531-7. Also that better references for influence of nutrients include: Vega Thurber et al. 2013 DOI 10.1111/gcb.12450, Zaneveld et al. 2016, DOI 10.1038/ncomms11833.

Thank you. We added the references on the impact of nutrients, but decided that further mentioning and discussion of bleaching would not be appropriate.

Lines 67-75: See comments above about the difference between ‘coral ecosphere’ and reef waters.

Changed accordingly. See above.

Line 73: What does reference 21 (the structure of bacterial communities in arctic ocean water) have to do with coral reefs or reef condition?

Reference was deleted.

Lines 79-80: The data do not address transmission or progression of SCTLD

 Sentence was deleted.

Methods – Lines 134-135 Insufficient information to allow someone to reproduce your bioinformatic analyses in QIIME2. See Reigel et al. 2020 DOI 10.1007/s00338-020-02006-5 for a good example.

We amended the methods in lines 119-138 and do think that they are now reproducible.

Results

Table S1: Confirming that station depth data are in meters (unit not specified)?

We added ‘m’ to the depth column in Table S1.

Line 216: “from stations from all three stations”

Corrected.

Figure 4: People with red/green color blindness will not be able to distinguish between the two numbers (suggest you chose a different color combination and/or augment by specifying left and right). Also, consider highlighting the few significant clusters by putting circles around them or light color block behind them.

We improved this figure by adding the original color scheme to the stations. We also highlight significant clusters now with red boxes and deleted the numbers at the nodes.

Line 239: Should reference Figure 5 not Figure 4?

Thanks, corrected.

Discussion

Lines 328-331: See Kline et al. 2006 MEPS 314: 119-125.

Thanks for this reference. We added it in line 310.

Lines 343-346: Reference? Also see Maher et al. 2020 DOI 10.3389/fevo.2020.555698

Added reference 20, Glasl et al 2019. While we see the value of the information reported in Maher et al., we think that discussing this reference here would be too speculative, given our limited dataset.

Lines 365-368: Highly speculative statement considering the shared taxa being discussed are at the order or family level (each of which contain lots of diversity at the genus, species, and strain level).

We completely agree that this is speculative, but we think that this is still a valuable comment which is mentioning.   

Lines 373-374: Interesting but speculative. If Thioglobaceae (PS007) is indicative of high sulfide, shouldn’t it correlate with low DO?

Thanks. They are usually aerobic sulfide-oxidizers and rely on high O2 and high sulfide concentrations. We think it is important to make these remarks, although we didn’t study the physiology of these organisms.

Conclusions – Line 416: You did not survey the coral ecosphere.

Deleted and rephrased.

Round 2

Reviewer 3 Report

I very much appreciate the response to my concerns and their modifications to the manuscript. I originally didn't have much concern for the actual science of this manuscript, mostly with the presentation and conclusions. Again, I appreciate the changes and clarification and taking my concerns seriously. I have no further issues with this manuscript.